# MicroCrackAttentionNeXt:
# Advancing Microcrack Detection in Wave Field Analysis Using Deep Neural Networks through Feature Visualization

## Abstract

Micro Crack detection using deep neural networks(DNNs) through an automated pipeline using wave fields interacting with the damaged areas is highly sought after. However, these high dimensional spatio-temporal crack data are limited, moreover these dataset have large dimension in the temporal domain. The dataset exhibits a pronounced class imbalance, with crack pixels accounting for an average of only 5% of the total pixels per sample. This severe imbalance presents a challenge for deep learning models when dealing with various microscale cracks, as the network tends to favor the majority class, often resulting in reduced detection accuracy. This study proposes an asymmetric encoder–decoder network with Adaptive Feature Reutilization Block for micro-crack detection. The impact of various activation and loss functions were examined through feature space visualisation using manifold discovery and analysis (MDA) algorithm. The optimized architecture and training methodology achieved an accuracy of 87.74%.

## 1 Introduction

Micro crack detection in materials is of significant importance due to the potential for catastrophic failures, which can lead to substantial financial losses and safety hazards in industries (Malekloo et al., 2022; Golewski, 2023). Detecting cracks in complex structures, like aircraft bodies or intricate machinery components, poses a substantial challenge using conventional methods like visual inspection or standard cameras, especially when dealing with complex geometries. The use of wave-based approaches for crack detection offers a powerful solution, as these methods allow for the analysis of structures that are not easily accessible or too complex to inspect manually.

Convolutional Neural Networks (CNNs) are especially good at processing spatial data due to their ability to capture local spatial correlations within an image (LeCun et al., 2015). Nevertheless, standard segmentation methods, such as vanilla architectures, demonstrate limited performance on this particular dataset, due to the complex spatio-temporal nature of the crack patterns. This becomes even more significant when the cracks represent a tiny minority in the dataset, leading to poor detection accuracy. This issue is enhanced when dealing with very small cracks, as they not only lead to data imbalance but may also cause minimal disruption in wave behaviour. In such cases, the waves may exhibit minimal changes, making it difficult for the model to detect the cracks accurately.

This challenge necessitates the development of a more tailored custom model. Our proposed MicroCrackAttentionNeXt is designed to overcome the limitations of vanilla models like UNet by incorporating enhanced spatial and temporal feature extraction. Unlike UNet Ronneberger et al. (2015), where the input and target share the same modality (image-to-image translation). Our model processes spatio-temporal input data and outputs spatial crack predictions, enabling it to handle more complex data while improving micro-scale detection accuracy. The asymmetric encoder-decoder structure, with attention layers is particularly effective as it focuses on capturing critical crack patterns rather than relying heavily on skip connections. The attention mechanism ensures that the model prioritizes the time steps when the waves interact with the cracks, improving detection precision. The DNNs capacity to recognise minute details and complex patterns in high dimensional data

is impacted by the activation functions used, which becomes crucial in the micro-scale setting where accuracy is much needed. Activation functions enhance the network's expressive power, enabling it to capture diverse features and representations.

Rectified Linear Unit (ReLU) Nair & Hinton (2010) and its variants are commonly used activation functions. ReLU introduces non-linearity by setting negative values to zero, allowing positive ones to pass unchanged, which aids in deep network training. The "dying ReLU" issue, where neurons become inactive, hampers learning Xu et al. (2015); He et al. (2015b). Variants like Leaky ReLU mitigate this by allowing small negative slopes. SELU (Scaled Exponential Linear Unit) Klambauer et al. (2017) scales outputs to maintain self-normalizing properties, keeping activations near zero mean and unit variance. GeLU (Gaussian Error Linear Unit) Hendrycks & Gimpel (2023) enhances representation learning by incorporating probabilistic elements, though it has higher computational complexity. ELU (Exponential Linear Unit) Clevert et al. (2016) improves learning dynamics but is computationally expensive. Various Loss functions have been proposed in the literature to combat class imbalance issues in the DNN model. The loss functions tested are: 1)Dice LossLin et al. (2018), 2)Focal LossLin et al. (2018), 3)Weighted Dice LossYeung et al. (2021) and, 4)Combined Weighted Dice LossJadon (2020).

These activation functions aim to strike a delicate balance between adaptability and computational efficiency, essential considerations in the micro-material domain, where capturing fine details is crucial for accurate crack detection. Empirical exploration and meticulous fine-tuning of these activation functions is imperative to identify the optimal choice that aligns with the distinctive characteristics of micro-material images. Ultimately, a nuanced and effective approach to crack detection in micro-materials relies on the thoughtful selection and optimization of activation functions within the CNN architecture.

The extent of the influence of different activations is difficult to determine against conventional metrics such as accuracy and F1 score. Hence, it is imperative to analyse the internal dynamics of the model. Methods like Principal Component Analysis, t-SNE van der Maaten & Hinton (2008) and UMAP McInnes et al. (2020) are used to analyse the higher dimensional feature maps of these blackbox models against the target. However, these methods provide little to no insight when used on segmentation problems. In this study, we use the recently proposed Manifold Discovery Analysis (MDA) Islam et al. (2023) to qualitatively assess the impacts of various activation functions. Moreover, through this, we were able to analyse the effects activations had on the feature maps of the model, allowing us to choose the best activation function for the given problem.

The primary contributions of this paper are:

- Introducing MicroCrackAttentionNeXt – an improvement over (Moreh et al., 2024).
- Qualitative Investigation of the impact of different network architectural choices, activations and loss functions in MicroCrackAttentionNeXt through Manifold Discovery and Analysis.

The paper's structure is outlined in the following manner: Section 2 encompasses a concise yet informative overview of relevant studies. Section 3 deals with the dataset used and the proposed methodology. The assessment of the performance of the proposed system and the results obtained are included in Section 4. Ultimately, concluding remarks and future works are presented in Section 5.

## 2 RELATED WORKS

In a number of areas, including materials science, aerospace, and infrastructure, where the existence of small fissures might jeopardise the structural integrity of materials, micro-crack detection is essential (Chen et al., 2024; Yuan et al., 2022; Retheesh et al., 2017). Conventional techniques for detecting microcracks are frequently labour-intensive and not very scalable. Deep learning, and Convolutional Neural Networks (CNNs) in particular, have become a potent and effective technique for microcrack detection automation in recent years(Su & Wang, 2020; Chen & Jahanshahi, 2017; Hamishebahar et al., 2022).

Tran et al. (2024) applied 1D Convolutional Neural Networks (1D CNNs) for structural damage detection, utilizing acceleration signals to detect cracks in numerical steel beam models. Their

approach showed high detection accuracy, comparable to more complex methods, by processing time-series data and extracting key features related to structural changes. While they focused on single-dimensional data, our research extends this by using multi-dimensional spatio-temporal data, which includes wave propagation across the material. This allows for more detailed analysis, capturing both spatial and temporal interactions crucial for detecting micro-cracks.

Jiang et al. (2022) combined 1D CNNs with Support Vector Machines (SVM) to enhance structural damage detection. 1D CNNs were used to localize damage, while SVMs focused on classifying the severity, benefiting from the strengths of both models in feature extraction and small-sample learning.

Barbosh et al. (2024) used Acoustic Emission (AE) waveforms and DenseNet to detect and localise the crack. The localisation was done to determine whether the crack was close to the sensor or far away. The cracks were also classified into severe and less severe cracks.

Moreover, Li et al. (2023) proposed a GM-ResNet-based approach to enhance crack detection, utilizing ResNet-34 as the foundational network. To address challenges in global and local information assimilation, a global attention mechanism was incorporated for optimized feature extraction. Recognizing limitations in ResNet-34, the fully connected layer was replaced with a Multilayer Fully Connected Neural Network (MFCNN), featuring multiple layers, including batch normalization and Leaky ReLU nonlinearity. This innovative substitution significantly improved the model's ability to capture complex data distributions and patterns, enhancing feature extraction and representation capabilities while preventing overfitting during training.

Wuttke et al. (2021) introduced a 1D-CNN-based model, SpAsE-Net, for detecting cracks in solid structures using wave field data. The model leverages sparse sensor data and the Dynamic Lattice Element Method (LEM) for wave propagation simulations. The network's architecture includes fully convolutional layers for spatial feature fusion and a predictor module using transposed convolutional layers and focal loss for crack localization. It achieves around 85% accuracy in detecting small cracks(>1 μm) and 97.4% accuracy in detecting large cracks(>4 μm) by tuning the focal loss parameters.

Moreh et al. (2022) present a DNN based method for detecting and localizing cracks in materials using spatio-temporal data. They introduce two CNN architectures: a SimpleCNN (SCNN) as a baseline model and a more complex Residual Network (ResNet18) encoder. SCNN and ResNet18 leverage 1D convolutions to extract temporal features from the wave data, followed by 2D convolutions for spatial feature extraction. Both models employ a decoder with transposed convolutional layers to upscale the encoded features and predict a binary mask indicating the crack locations. The models were evaluated on simulated wave propagation data, where cracks ranging from 0.4 to 12.8 μm in size. ResNet18 outperformed SCNN and achieved a precision of 0.92, recall of 0.719 and a DSC of 0.744.

Moreh et al. (2024) explores the use of DNN for automated crack detection in structures using seismic wave signals. The authors improve on previous asymmetric encoder-decoder models by experimenting with different encoder backbones and decoder layers. The best combination was found to be the 1D-DenseNet encoder and the Transpose Convolutions as decoders. The proposed model achieved an accuracy of 83.68% with a total parameter count of 1.393 million.

This study builds upon the foundational contributions of Wuttke et al. (2021) and Moreh et al. (2024), extending their methodologies to a broader scope. The existing body of work in this field remains relatively sparse, with few studies addressing crack segmentation through the specific approach employed in this research.

## 3 METHOD

Our proposed work targets the detection of microcracks across various sizes and locations within seismic wave field numerical data. For this purpose, our MicroCracksAttentionNeXt extracts crucial signals from the data to identify and detect those cracks. This is done by first learning the temporal representations, followed by spatial representations. This encoded data is then passed through the decoder to achieve semantic spatial segmentation.

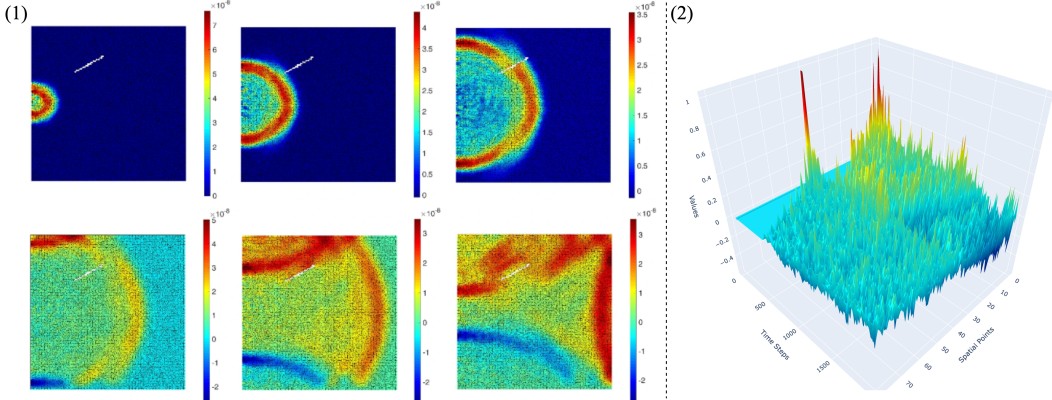

Figure 1: 1) The 6 frames (100 time-steps interval, from left to right) of a displacement() wave propagation inside the defined plate with cracks(Wuttke et al., 2021). 2) Visualization of a data instance.

In this section, we describe the seismic wave data followed by the architecture of the proposed MicroCrackAttentionNeXt model and, subsequently, the training procedure used.

## 3.1 WAVE FIELD DATA

In this study, it is crucial to understand that it deals with numerical data produced by geo-science experts at our group (Wuttke et al., 2021). For someone without expertise in this field, interpreting the data is nearly impossible, as the data is purely numerical and provides no visual insights. The goal of this research is to demonstrate that it is indeed possible to extract patterns from such data. Research in this area is still in its infancy, making it an important achievement to prove that machine learning can process such data and segment fractures. We are also working on generating a new dataset that better represents reality, where real-world data can be evaluated using our model.

The wave field dataset utilized in this work (Wuttke et al., 2021), while effective for crack detection, presents some limitations in terms of data dimensionality. These datasets are characterized by large temporal dimensions, which increases the complexity of data processing and model training. The dataset consists of homogeneous 2D plates, where each plate is modelled with lattice particles that share consistent properties, such as density and Young's Modulus. The modeling of structural systems is achieved using Voronoi-Delaunay meshing algorithms within the Lattice Element Method (LEM). Lattice nodes, representing unit cell centers, are connected by beams capable of handling normal forces ($N$), shear forces ($V$), and bending moments ($M$). If the strain energy $U_e$ in an element exceeds a predefined threshold $U_{\text{th}}$, the element undergoes stiffness reduction or removal, simulating failure states. To simulate wave propagation through the material, an external force of 1000 N is applied at the midpoint of the left boundary, ensuring that the waves propagate across the entire plate, interacting with both non-crack and crack regions. The resulting displacements in both the x- and y-directions are recorded over 2000 time steps, capturing detailed temporal changes in the wave field. These displacements are measured by a $9 \times 9$ (81) sensor grid uniformly distributed across the material, resulting in a wave field dataset with dimensions of 2 x 81 × 2000. Figure 1 shows a sample of the input dataset. This approach provides spatio-temporal data that captures the interaction between the propagating waves and the cracks, allowing for in-depth analysis of crack detection model performance.

A major challenge arises from the severe class imbalance present in the dataset. On average, only 5% of the total pixels represent cracks, with the remaining majority belonging to intact, non-crack regions. This imbalance poses a substantial obstacle for deep learning models, which are prone to bias toward the majority class. As a result, the models tend to predict non-crack regions more frequently, leading to suboptimal detection accuracy for the minority class (crack regions). Addressing this issue requires careful design of the model and training process to ensure that the network can effectively learn from the minority class, and accurately identify crack regions without being overpowered by the majority class imbalance.

## 3.2 MICROCRACKSATTENTIONNEXT MODEL ARCHITECTURE

MicrocrackAttentionNeXt, shown in Figure 2, is an asymmetric encoder-decoder network. The input to the model is a tensor with shape $\mathbf{X} \in \mathbb{R}^{C_{\text{in}} \times T \times S}$, where $C_{\text{in}} = 2$ represents the input channels corresponding to the $x$ and $y$ components of wave data, $T = 2000$ is the temporal dimension, and $S = 81$ corresponds to the spatial dimension, which is a flattened $9 \times 9$ sensor grid. To reduce computational complexity and focus on salient temporal features, the network uses an initial max pooling layer with a kernel size of $(4, 1)$. This layer transforms the input tensor $\mathbf{X}$ to $\mathbf{X}_1 \in \mathbb{R}^{2 \times 500 \times 81}$ by downsampling the temporal dimension from 2000 to $T_1 = 500$. This reduction is crucial as it reduces the amount of data the subsequent layers need to process.

The encoder is composed of four convolutional blocks, each designed to progressively extract higher-level features from the input data. The first convolutional block applies two convolutional layers with kernel sizes $(3, 1)$ and padding $(1, 0)$, which maintain the spatial dimensions while expanding the channel dimension from 2 to 16. These layers are followed by batch normalization and activation functions, introducing non-linearity. A Squeeze-and-Excitation (SE) module is then applied, which recalibrates channel-wise feature responses by modelling interdependencies between channels. This module enhances the representational power of the network by allowing it to focus on the most informative features. Following the first convolutional block, a max pooling layer with a kernel size of $(2, 1)$ further reduces the temporal dimension from 500 to $T_2 = 250$. Group normalization is applied to the data, normalizing across channels and improving convergence during training. An AttentionLayer computes self-attention over the temporal and spatial dimensions, enabling the network to weigh different parts of the input differently. This attention mechanism is essential for focusing on relevant features and capturing dependencies across the data. A residual connection adds the attention output back to the original input, facilitating better gradient flow and mitigating issues such as vanishing gradients (Raghu et al., 2022; He et al., 2015a).

This pattern repeats in the subsequent convolutional blocks, with each block increasing the number of channels (from 16 to 32, 32 to 64, and 64 to 128) and further reducing the temporal dimension (from 250 to 125, 125 to 62, and 62 to 31) through additional pooling layers. The consistent use of $(3, 1)$ kernels ensures effective temporal feature extraction while preserving spatial dimensions. SE modules and attention mechanisms are integrated throughout. Feature maps are upsampled and reintroduced to the Conv1 block through a Self-Attention Module (SAM)-inspired mechanism, enabling the decoder backbone to reuse features and increase model performance. The feedback mechanism employs bilinear interpolation for resizing and utilizes Conv2D layers to selectively regulate the features propagated back into the network giving it the alias of Adaptive Feature Reutilization block.

At the bottleneck of the network, a convolutional layer with a large kernel size of $(31, 1)$ is employed, covering the entire temporal dimension $T_5 = 31$. This layer transforms the tensor to $\mathbf{X}_{\text{bottleneck}} \in \mathbb{R}^{B \times 128 \times 1 \times 81}$, capturing long-range temporal dependencies and encapsulating high-level temporal information into a compact form. Batch normalization and activation are applied to maintain training stability and introduce non-linearity.

The decoder begins by reshaping this bottleneck tensor into a spatial grid $\mathbf{X}_{\text{reshaped}} \in \mathbb{R}^{128 \times 9 \times 9}$, reorganizing the data for spatial processing. A point-wise convolution reduces the channel dimension from 128 to 16, preparing the data for upsampling. The network then uses transposed convolutional layers to reconstruct the spatial dimensions progressively. The first transposed convolution upsamples the spatial dimensions from $9 \times 9$ to $18 \times 18$ and reduces the channel dimension from 16 to 8. The second transposed convolution further upsamples the dimensions to $36 \times 36$, maintaining the channel count at 8. Each transposed convolution is followed by batch normalization to ensure stable learning and effective non-linear transformations.

Finally, a point-wise convolution reduces the channel dimension from 8 to 1, and a sigmoid activation function scales the output to the range $[0, 1]$. The output tensor $\mathbf{Y} \in \mathbb{R}^{1 \times 36 \times 36}$ represents the reconstructed spatial data, which is then flattened into a vector $\mathbf{Y}_{\text{flat}} \in \mathbb{R}^{1296}$ (since $36 \times 36 = 1296$), making it suitable for downstream tasks. Figure 2 shows the proposed model architecture. The architectural choices in MicrocrackAttentionNeXt are designed to balance feature extraction capability and computational efficiency. The initial temporal downsampling reduces the data size, allowing the network to process longer sequences without excessive computational overhead. The 1D convolutional blocks with increasing channel dimensions enable the extraction of hierarchical features in the temporal domain, without mixing the spatial component. We found that learning temporal and

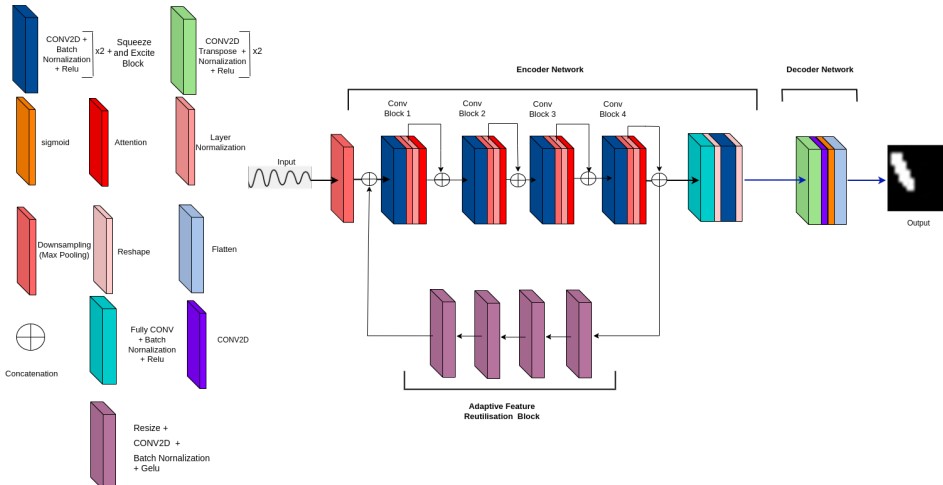

Figure 2: MicroCrackAttentionNeXt model architecture.

spatial components separately enables the model to learn better representations, and it is much more computationally efficient. The Squeeze-and-Excitation layers optimize the network's focus on informative channels, improving feature quality. Using a large kernel size in the bottleneck layer is an intentional choice to capture long-range temporal dependencies, which are important in sequences where connected events are separated by large time steps. The reshaping and upsampling in the decoder reconstruct the spatial dimensions effectively, ensuring that the high-level features extracted by the encoder are used to generate outputs.

## 3.3 TRAINING PROCEDURE

The model was trained using the Adam optimizerKingma & Ba (2017) with a learning rate of 0.001 for a total of 50 epochs. Multiple experiments were run on different activation functions and loss metrics. The experiments involved evaluating four different activation functions against four loss metrics, resulting in a total of 16 experiments. The activation functions and loss metrics are outlined below.

### 3.3.1 ACTIVATION FUNCTIONS

Activation functions are used to introduce non-linearity within neural networks, each offering different advantages for a DL model. The Rectified Linear Unit (ReLU) is defined as $\text{ReLU}(x) = \max(0, x)$, outputting the input if positive and zero otherwise, thus avoiding vanishing gradient issues. The Scaled Exponential Linear Unit (SELU) normalizes outputs automatically, scaling negative inputs with an exponential function and multiplying positive inputs by a fixed constant, where $\lambda = 1.0507$ and $\alpha = 1.67326$. The Gaussian Error Linear Unit (GELU) employs the Gaussian cumulative distribution function, $\Phi(x) = \frac{1}{2}\left[1 + \text{erf}\left(\frac{x}{\sqrt{2}}\right)\right]$, to probabilistically weigh input significance. GELU smoothly blends linear and non-linear behavior, making it more flexible in capturing complex patterns The Exponential Linear Unit (ELU) applies $\text{ELU}(x) = x$ for positive inputs and $\alpha(e^x - 1)$ for negatives, mitigating vanishing gradients more effectively than ReLU, and accelerating convergence, with $\alpha$ typically set to 1. Each function enhances network performance through tailored non-linear transformations.

### 3.3.2 LOSS FUNCTIONS

1. **Dice Loss**:
   Dice Loss is based on the Dice coefficient and is commonly used for segmentation tasks. It measures the overlap between the predicted and true labels, focusing on improving perfor-

mance for imbalanced datasets.

$$\text{Dice Loss} = 1 - \frac{2|X \cap Y|}{|X| + |Y|} \tag{1}$$

where $X$ and $Y$ are the predicted and true sets, respectively.

2. **Focal Loss**:
Focal Loss is designed to address class imbalance by down-weighting the loss assigned to well-classified examples, making the model focus more on hard-to-classify instances.

$$\text{Focal Loss}(p_t) = -\alpha(1 - p_t)^\gamma \log(p_t) \tag{2}$$

where $p_t$ is the predicted probability, $\alpha$ is a weighting factor, and $\gamma$ is a focusing parameter.

3. **Weighted Dice Loss**:
Weighted Dice Loss is a variation of Dice Loss that assigns different weights to different classes, enhancing performance on datasets with imbalanced class distributions by penalizing certain classes more.

$$\text{Weighted Dice Loss} = 1 - \frac{2 \sum w_i x_i y_i}{\sum w_i x_i^2 + \sum w_i y_i^2} \tag{3}$$

where $w_i$ is the weight assigned to class $i$, and $x_i$, $y_i$ are the predicted and true values for class $i$.

4. **Combined Weighted Dice Loss**:
This is a hybrid loss that combines Weighted Dice Loss and CrossEntropy Loss, allowing the model to balance overall performance while addressing class imbalances by tuning the contribution of each component.

$$\text{CWDL} = \alpha \cdot \text{WDL} + (1 - \alpha) \cdot \text{CrossEntropy Loss} \tag{4}$$

where CWDL is Combined Weighted Dice Loss, WDL is Weighted Dice Loss and, $\alpha$ is a weighting factor to balance the two loss components.

We found the combination of Combined Weighted Dice Loss and GeLU to be the best performing. The combined weighted dice loss performed the best across all the activations. However, we found that we were able to squeeze more accuracy through the GeLU function.

## 3.4 EVALUATION METRICS

For the evaluation part, we utilized the same metrics as in Moreh et al. (2024), namely Dice Similarity Coefficient (DSC) and accuracy which frequently employed to evaluate the performance of models. The DSC measures the overlap between predicted and actual results, particularly in segmentation tasks. Its mathematical formulation is given by:

$$\text{DSC} = \frac{2 \cdot \text{TP}}{2 \cdot \text{TP} + \text{FP} + \text{FN}} \tag{5}$$

Accuracy measures the overall correctness of the predictions by calculating the proportion of true results, both positive and negative, over the total number of cases, given by:

$$\text{Accuracy} = \frac{\text{TP} + \text{TN}}{\text{TP} + \text{TN} + \text{FP} + \text{FN}} \tag{6}$$

## 4 RESULTS & DISCUSSION

### 4.1 MDA ANALYSIS

Manifold Discovery and Analysis (MDA) helps visualize the higher dimensional manifolds formed by the intermediate layers of the model in lower dimension (Islam et al., 2023). These plots help visualise the learned features in the $\ell^{\text{th}}$ layer with respect to the output manifold. Unlike methods like t-SNE and UMAP, which only work on classification tasks, MDA works on regression tasks,

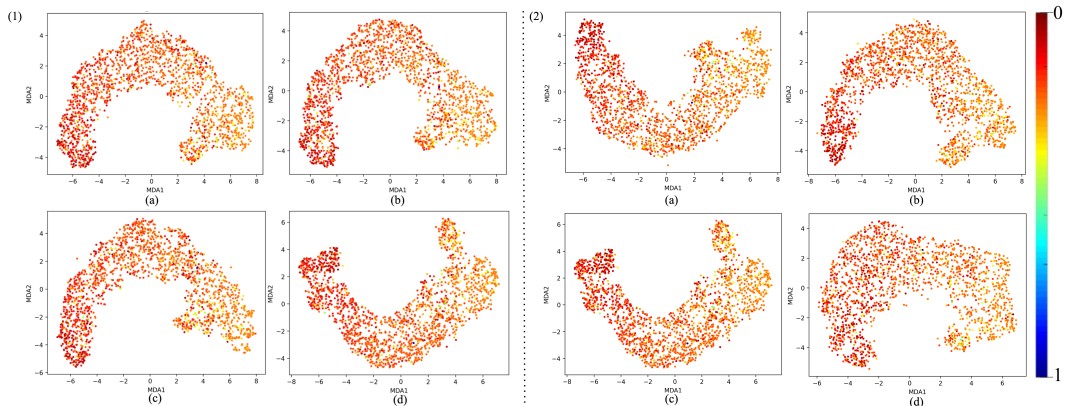

Figure 3: 1) MDA visualizations of layers using Gelu activation and Dice loss, shown for a) Layer 22, b) Layer 25, c) Layer 34, and d) Layer 64 and, 2) MDA visualization of Layer 64 utilizing different activation functions: a) ELU, b) ReLU, c) GELU, and d) SELU.

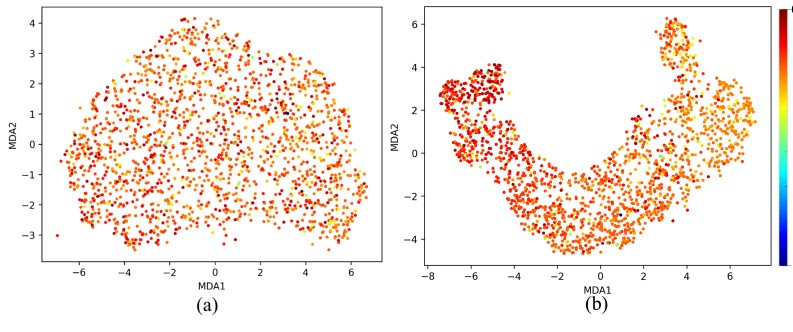

Figure 4: MDA visualization of Layer 64 comparing a) Untrained Model and b) Trained Model.

where the output manifold can have a complex shape. MDA also preserves the geodesic distances between higher dimensional feature points, preserving both local and global structure.

In a nutshell, MDA works as follows: First, distance is computed between the estimated outputs of the DNN, from this distance the farthest point is chosen to construct the boundary of the output manifold. All the points are sorted w.r.t the farthest point in k bins using optimal histogram bin count. These bins become the labels that will be used in the second step. Second, the high dimensional features from an intermediate layer are projected to the manifold using the Bayesian manifold projection (BMP) approach. BMP computes a posterior distribution over the low-dimensional space by combining the prior (based on pseudo-labels and manifold structure) with a likelihood (based on the observed data). Finally, a DNN trained on predicting the location of uncertain Bayesian points on a 2D embedding space is used to visualise the results. The plots are assessed qualitatively on the following points:

- Feature Separation and Continuity: The MDA visualization shows a curved shape, indicating that the features extracted from the neural network follow a smooth continuum along the manifold. This suggests that the neural network is capturing meaningful information.

- Color Gradient: A spectrum of gradients is shown, implying that the model has learned to separate different features.

The MDA plot for the untrained model shows a relatively disorganized and diffuse clustering of points. This suggests that the feature representations at Layer 64 are not yet structured in a meaningful way to distinguish between patterns within the dataset. The absence of clear separation or distinct clustering patterns indicates that the untrained model has not yet learned to capture the un-

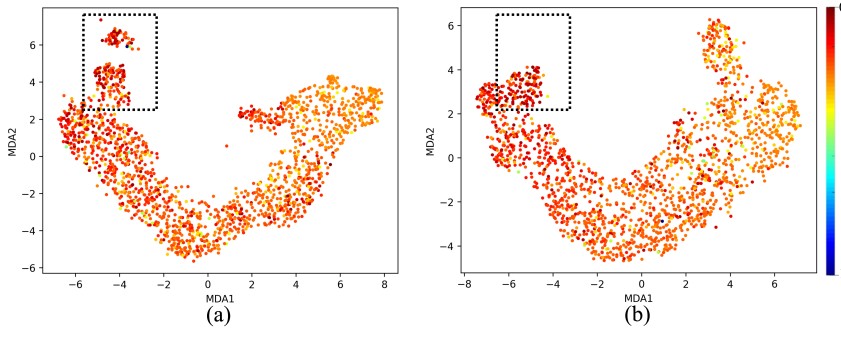

Figure 5: MDA visualization comparing a) 1D-Densenet Moreh et al. (2024) and b) Our proposed model - MicroCrackAttentionNeXt. The highlighted region in black indicates the region where the cluster is broken in 1D-Densenet. In contrast, the same region in MicroCrackAttentionNeXt shows coherency implying that the MicroCrackAttentionNeXt learned good feature representations for microcracks.

derlying structure of the data, which is expected at the initial stages of training. At this stage, the network's representations are largely random, as it has not yet learned the task-specific features. The spread-out nature of the points highlights that the model is treating all inputs similarly, without any differentiation based on the features it should detect. In contrast, the MDA plot for the trained model reveals a much more structured and organized distribution. The $\ell^{th}$ layer of a well-trained model shows the cluster with a smooth arch-like structure in fig 5 and a gradient of colors differentiating the two output extremums, in our case red representing 0 and Blue representing 1. The analysis of various layer depths and activation functions in MicroCrackAttentionNeXt reveals a clear pattern in the network's ability to form distinguishable manifold curves. Fig 3(1) visualises the manifold at different layers of the network. All the layers show consistent and smooth arch-like shapes. This implies that all the layers have learned good representations. In fig 3(2) effects of various activations are plotted, we see that ELU shows more spread out cluster especially toward the light colors (towards crack class), ReLU shows a good arch like structure with tightly packed dots of red color (no crack class). Still, the light color dots are much more incoherent and less compact. SeLU shows a poorly defined structure compared to all other activations. This is also reflected in the results table where SeLU performs worse in all cases. This behaviour of SeLU can be attributed to its self-normalising property, as it forces all outputs to behave similarly, "dampening" the importance of smaller, rare patterns. This makes the model less discriminative, making the plot less defined and incoherent. GeLU in terms of cluster shape and compactness is similar to ReLU, which should be the case as GeLU is smoothed version of ReLU. We also observe very similar performance in the results table. Among all the activations GeLU performed the best, this fact also reflects on the MDA plot where the cluster is very smooth and the lighter points are more compactly packed relative to other activation functions. Its smooth probabilistic gating mechanism helps in finely controlling how information is passed through the network, allowing the model to focus more on the minority class. Fig 5 shows MDA visualisation of 1-D Densenet proposed in Moreh et al. (2024) is compared with the proposed MicroCrackAttentionNeXt model. One thing to note is the absence of full spectrum of colors in the plot. This is mainly attributed to the severe class imbalance in the data. This class imbalance leads to very few values in the feature map strongly correlating to the strong value of predicting the crack class. This is further aggravated by the dimensionality reduction, which renders even fewer points corresponding to higher confidence value. Hence we see often only one point representing Blue color. The proposed MicroCrackAttentionNeXt achieved a DSC of 0.91. Table 1 shows the comparison of different loss functions used to train MicroCrackAttentionNeXt.

Table 1: Comparison of accuracies using different loss functions for multiple crack sizes. FL: Focal Loss, DL: Dice Loss, WDL: Weighted Dice Loss, CWDL: Combined Weighted Dice Loss

| Activation function | Loss function | > 0 μm | > 1 μm | > 2 μm | > 3 μm | > 4 μm |
|---|---|---|---|---|---|---|
| GeLU | FL | 0.8275 | 0.8612 | 0.9354 | 0.9501 | 0.9541 |
| | DL | 0.8633 | 0.9012 | 0.9585 | 0.9701 | 0.9802 |
| | WDL | 0.8381 | 0.8798 | 0.9415 | 0.9670 | 0.9793 |
| | CWDL | **0.8774** | **0.9211** | **0.9814** | **0.9808** | **0.9848** |
| ReLU | FL | 0.8252 | 0.8632 | 0.9456 | 0.9701 | 0.9802 |
| | DL | 0.8553 | 0.8902 | 0.9646 | 0.9770 | 0.9829 |
| | WDL | 0.8213 | 0.8687 | 0.9293 | 0.9524 | 0.9703 |
| | CWDL | 0.8678 | 0.9134 | 0.9673 | 0.9808 | 0.9866 |
| ELU | FL | 0.8313 | 0.8797 | 0.9558 | 0.9839 | 0.9911 |
| | DL | 0.8502 | 0.9011 | 0.9673 | 0.9831 | 0.9884 |
| | WDL | 0.8563 | 0.9034 | 0.9605 | 0.9739 | 0.9829 |
| | CWDL | 0.8515 | 0.9041 | 0.9673 | 0.9847 | **0.9920** |
| SeLU | FL | 0.8206 | 0.8671 | 0.9503 | 0.9793 | 0.9902 |
| | DL | 0.8412 | 0.8993 | 0.9707 | **0.9870** | 0.9893 |
| | WDL | 0.8201 | 0.8664 | 0.9307 | 0.9555 | 0.9712 |
| | CWDL | 0.8443 | 0.8910 | 0.9625 | 0.9854 | 0.9929 |

## 5 CONCLUSION AND FUTURE WORK

### 5.1 CONCLUSION

Through this study, we have demonstrated the effectiveness of feature visualization in designing MicroCrackAttentionNeXt, by carefully optimizing the architecture, leveraging the right activation function and loss. This architecture also utilizes multiple 1D-CNN layers for feature extraction, significantly reducing the training time. These are followed by folded layers that merge spatial and temporal features, along with a prediction module for semantic segmentation. The dataset used are spatio-temporal in nature and represent the behavior of wave propagation, where waves interact with the cracks, leading to disruptions in their patterns and altered behavior in the presence of cracks. The model is capable of segmenting the microcracks, helping to determine their spatial locations in the material. The qualitative examination of the activation functions using the Manifold Discovery and Analysis (MDA) algorithm allowed the evaluation of impact of different activation and loss functions on the model's performance. The proposed model and 1D-Densenet were analyzed using the MDA plots. It was observed that manifold of the proposed model was more compact with a much more smoother arc than 1D-Densenet. With the optimized selection of activation and loss functions, an accuracy of 87.74% was achieved.

### 5.2 FUTURE WORK

In future efforts to improve microcrack detection models, two primary strategies can be pursued: expanding datasets and refining model architectures. The dataset used, presents a challenge due to severe class imbalance, which requires more advanced techniques for data generation and augmentation to mitigate the bias introduced. Moreover, the segmentation output suffers from low resolution, and without appropriate upscaling techniques, critical details may be lost. To address this, in the future works, we propose incorporating a super-resolution GAN approach to enhance the resolution of the segmentation outputs. While the encoder architecture performs optimally, further changes are necessary in the decoder section of the segmentation model to achieve improved results and maintain consistency with the high-quality input features. To enhance the encoder's ability to capture long-range dependencies, state space model can be used, particularity integrating the recently proposed Mamba architecture. This adjustment would improve the model's ability to handle complex spatial relationships, thereby strengthening feature extraction and contributing to overall performance gains in the segmentation task.

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
