# MicroCrackAttentionNeXt:

# Advancing Microcrack Detection in Wave Field Analysis Using Deep Neural Networks through Feature Visualization

## Abstract

Micro Crack detection using deep neural networks(DNNs) through an automated pipeline using wave fields interacting with the damaged areas is highly sought after. However, these high dimensional spatio-temporal crack data are limited, moreover these dataset have large dimension in the temporal domain. The dataset exhibits a pronounced class imbalance, with crack pixels accounting for an average of only 5% of the total pixels per sample. This severe imbalance presents a challenge for deep learning models when dealing with various microscale cracks, as the network tends to favor the majority class, often resulting in reduced detection accuracy. This study proposes an asymmetric encoder–decoder network with Adaptive Feature Reutilization Block for micro-crack detection. The impact of various activation and loss functions were examined through feature space visualisation using manifold discovery and analysis (MDA) algorithm. The optimized architecture and training methodology achieved an accuracy of 87.74%.

## Supplementary Materials

### Architectural Choices

The baseline model architecture, MicroCrackAttentionNeXt, as illustrated in Figure 6, serves as the foundation for evaluating the impact of architectural adjustments. Each modification's effect on accuracy across crack sizes is analyzed through corresponding accuracy vs. crack size graphs.

### M1: Vanilla (Baseline)

The performance of the vanilla model is depicted in Figure 1, which plots accuracy against architectural modifications for different crack sizes. This has a simple convolution based encoder. It serves as the baseline for further investigation of impact of various architecture changes. The vanilla encoder shows satisfactory performance across various crack sizes.

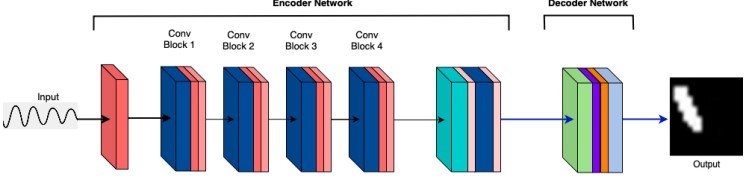

Figure 1: M1: Vanilla (Baseline) model architecture.

**M2: Adding Adaptive Feature Reuse Block:** The adaptive feature reuse block shows marginal improvement in accuracies across crack sizes. This is due to the increased discriminative nature of the block because of the sigmoid layer in the block's last layer.

ATTENTION MECHANISM PLACEMENT

**M3: Self Attention Layers:** Self attention after convolution blocks do not affect the accuracies of the model. This is after the skip connections.

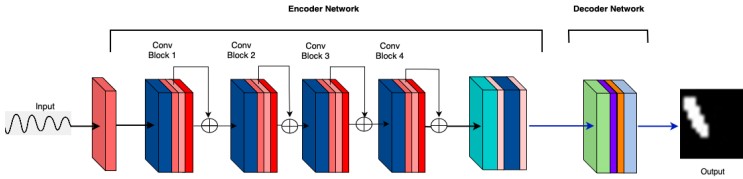

Figure 2: M3: Self Attention layer after convolution block.

**M4: Attention before Max Pooling in the First Layer:** Figure 3 illustrates the accuracy trends for attention applied before pooling. At Epoch 50, the performance remains suboptimal across different crack sizes. This is due to the information bottleneck caused the attention layers.

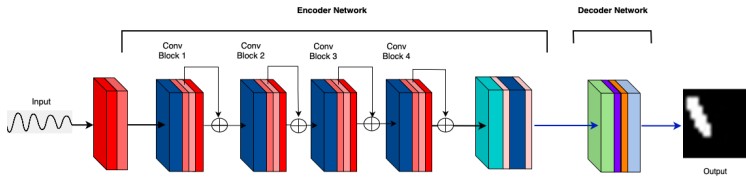

Figure 3: M4: Attention before Max Pooling in the first layer.

**M5: Attention before Max Pooling (Prolonged training):** Prolonged training to Epoch 100, results in even poorer performance across all crack sizes, suggesting that early attention causes severe information bottlenecks.

**M6: Attention after Max Pooling:** Performance remains suboptimal after attention after max pooling configuration. Figure 4 shows the performance of the Attention after Max Pooling. This is also caused by the bottleneck issue.

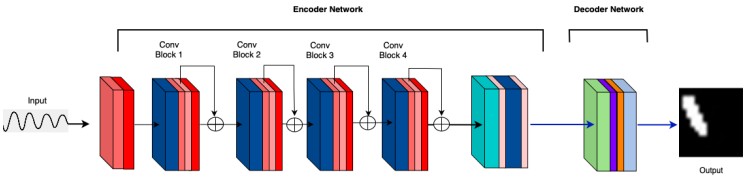

Figure 4: M6: Attention after Max Pooling in the first layer.

POOLING VARIANTS

**M7: 4x1 Max Pooling and Average Pooling Hybrid:** Using max pooling for the first layer and average pooling in the subsequent layers. The graph indicates virtually no change in the accuracy. This implies that average pooling is not very different from max pooling.

**M8: All Average Pooling:** The results for this configuration, reveal slightly higher accuracy for crack sizes $> 1\,\mu$m and $> 2\,\mu$m. Otherwise, the graph indicates virtually no change in the accuracy.

**M9: Convolutional Pooling Layers:** Figure 5 compares the performance of convolutional pooling layers at Epoch 50. The results demonstrate decreased accuracy across all crack sizes, with more dips for smaller cracks. This highlights the advantage of max pooling over convolutional downsampling spatially, as it ensures the preservation of dominant features within each pooling region

without introducing additional learnable parameters. By focusing on the maximum value, max pooling effectively captures critical localized details, such as the subtle variations indicative of smaller cracks, which are often diluted by the averaging effect of convolutional downsampling.

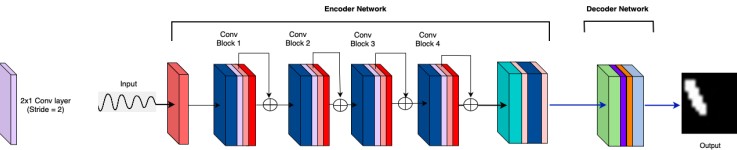

Figure 5: M9: Convolutional Pooling layers.

**M10: Convolutional Pooling Layers (Prolonged training):** Prolonged training to Epoch 100, enhances performance slightly, emphasizing the trainable pooling's ability to adaptively refine features. Overall, the deviation remains within a maximum of 2% for all crack sizes, indicating no major improvements to the model.

This highlights the advantage of max pooling over convolutional downsampling spatially, as it ensures the preservation of dominant features within each pooling region without introducing additional learnable parameters. By focusing on the maximum value, max pooling effectively captures critical localized details, such as the subtle variations indicative of smaller cracks, which are often diluted by the averaging effect of convolutional downsampling.

CONSECUTIVE ATTENTION LAYERS IN THE ENCODER

**M11: Two Attention Layers (Epoch 50):** The graph in Figure 6 reveals improved accuracy for smaller crack sizes due to the enhanced focus provided by stacked attention layers. The benefits are more pronounced at finer resolutions.

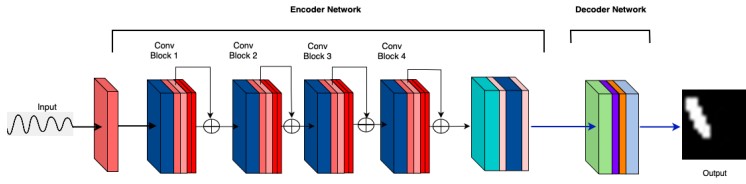

Figure 6: M11: Two Attention layers.

**M12: Two Attention Layers (Prolonged Training):** Prolonged training of this configuration, results in improvement in accuracies, however it still remains same across the board.

**M13: MicroCrackAttentionNext with single FeatureReuse:** The model shown in Figure 9 demonstrates strong performance across all thresholds. It achieves an overall accuracy of 85.74% for all cracks ($\geq 0$) and progressively higher accuracies for larger cracks, reaching 97.48% for cracks $> 4\,\mu$m. The single feature reuse mechanism effectively balances the retention of critical features, aiding in accurate detection of both small and large cracks.

**M14: MicroCrackAttentionNext with dual FeatureReuse:** Incorporating feature reuse twice results in slightly lower accuracy compared to the single feature reuse model. The accuracy for all cracks ($\geq 0$) drops to 83.24%, with corresponding reductions observed across all thresholds. For cracks $> 4\,\mu$m, the model achieves 97.2%, showing its capability for detecting larger cracks but indicating potential redundancy or noise in the reused features, which may hinder performance on smaller cracks.

**M15: MicroCrackAttentionNext with dual FeatureReuse and Extended Training:** Extending the training to 100 epochs with two feature reuse steps does not significantly improve performance. The overall accuracy further drops to 82.27%, and the accuracy for larger cracks is $> 4\,\mu$m is 96.3%. The results suggest that prolonged training does not compensate for the challenges introduced by additional feature reuse, particularly in handling smaller cracks effectively.

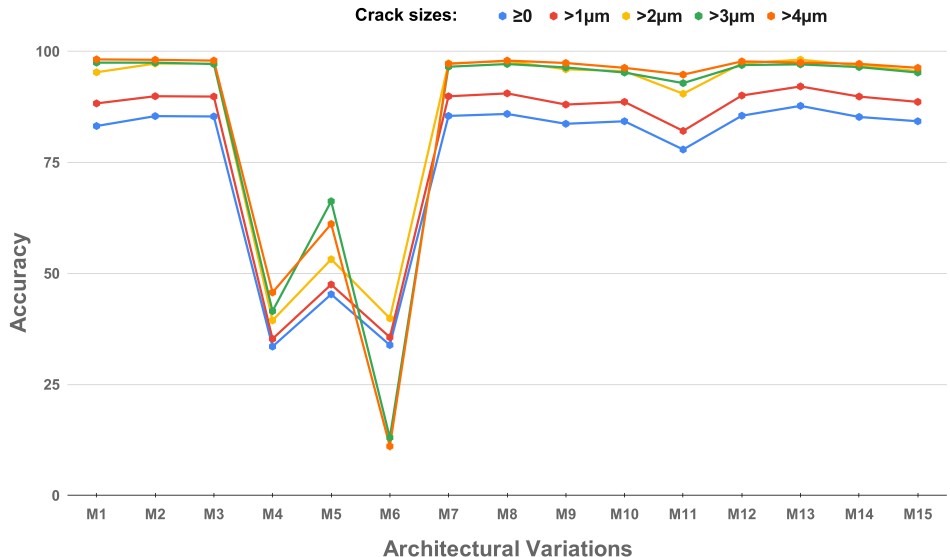

Figure 7: Performance comparison of architectural configurations for crack detection: **M1**: Baseline (MicroCrackAttentionNext 50E), **M2**: Attention after max pooling (50 epochs), **M3**: Attention after max pooling (100 epochs), **M4**: Attention before max pooling (50 epochs), **M5**: Hybrid pooling (4x1 max + average, 50 epochs), **M6**: All average pooling (50 epochs), **M7**: Convolutional pooling (50 epochs), **M8**: Convolutional pooling (100 epochs), **M9**: Two attention layers (50 epochs), **M10**: Two attention layers (100 epochs), **M11**: Single FeatureReuse (50 epochs), **M12**: Dual FeatureReuse (50 epochs), **M13**: Dual FeatureReuse (100 epochs), **M14**: Without attention layer.

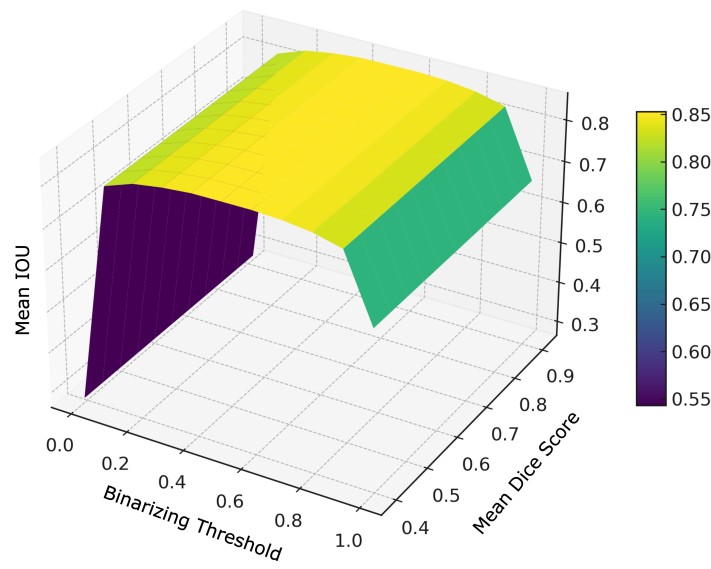

Figure 8: Surface plot consisting of binarising threshold with Mean IOU and Mean Dice Score.

## MORE RESULTS

Pixel-level predictions were binarized using a threshold of 0.5, where values exceeding this threshold indicated the presence of a crack. Prediction accuracy was assessed based on the Intersection over

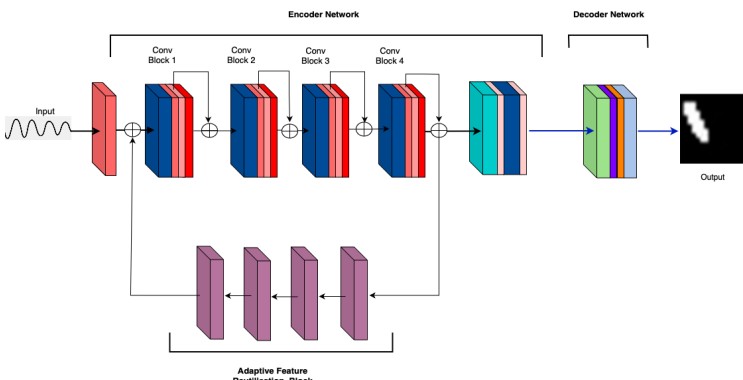

Figure 9: M13: MicroCrackAttentionNext with single FeatureReuse model architecture.

Union (IoU) metric, with a threshold of 0.5 selected to determine accurate crack identification. This threshold ensures a balance between minimizing false positives and maintaining practical accuracy.

Figure 10 demonstrates the relationship between varying IoU thresholds and overall accuracy. An IoU value approaching 1 requires near-perfect pixel classification but results in a significant accuracy drop due to its stringent nature. Conversely, an IoU value near 0 allows for minimal correct pixel identification but fails to filter out false positives. The chosen IoU threshold of 0.5 provides a practical compromise, achieving reasonable accuracy while maintaining meaningful crack identification.

Figure 11 explores the influence of both IoU and binarizing thresholds on accuracy, holding one threshold constant at 0.5. Variations in the binarizing threshold showed minimal impact on accuracy, except at extreme values (near 0 or 1), indicating strong confidence in the model's pixel-level predictions.

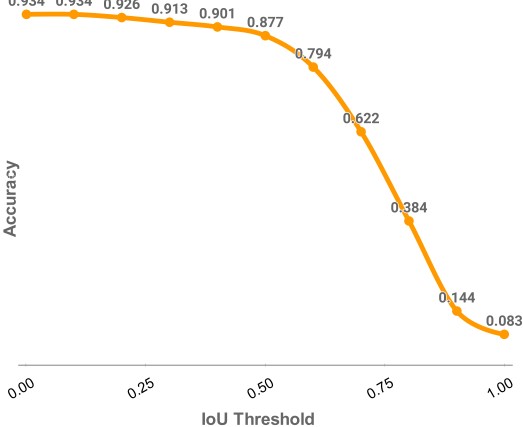

Figure 10: Effect of IOU threshold on accuracy for bin threshold at 0.5 and inclusive of all crack size.

### COMPARISON WITH SIMILAR WORKS

The uniqueness of this work lies in the dataset used and the applied deep learning algorithms. The data, which were exclusively collected at our university, consist of purely numerical values, distinguishing them from the image data commonly used in the literature. To the best of our knowledge, no deep learning models have been trained on this type of numerical data. Our goal is to develop models that not only detect cracks but also precisely segment them.

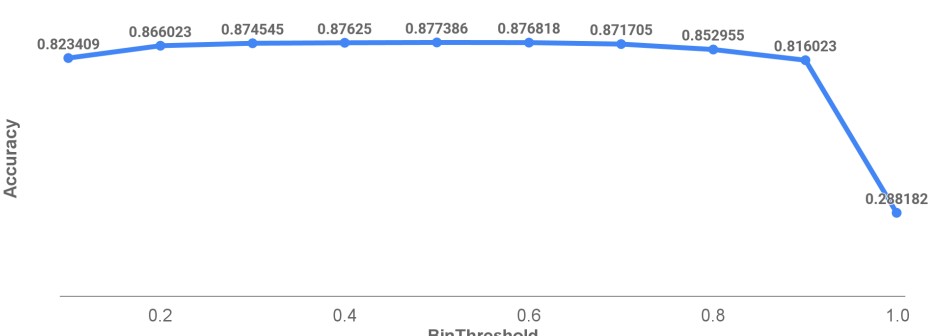

Figure 11: Binarising threshold when IoU threshold is 0.5.

Table 1: Comparison against previous works on the same dataset.

| Literature | Accuracy | IoU | DSC | Precision | Recall |
|---|---|---|---|---|---|
| 1D-CNN-Resize & Conv | 0.81875 | 0.756758 | 0.861539 | 0.869824 | 0.853411 |
| 1D-ResNet-Resize & Conv | 0.826875 | 0.772105 | 0.871399 | 0.883311 | 0.859803 |
| 1D-DenseNet-no decoder | 0.803125 | 0.717579 | 0.83557 | 0.83317 | 0.837984 |
| 1D-DenseNet-TConv | 0.836875 | 0.76014 | 0.863727 | 0.875395 | 0.852366 |
| 1D-DenseNet-Subpixel | 0.83625 | 0.757815 | 0.862224 | 0.880307 | 0.844868 |
| 1D-DenseNet-Resize & Conv | 0.834375 | 0.768578 | 0.869148 | 0.881086 | 0.857529 |
| **Ours** | **0.877** | **0.852** | **0.914** | **0.8601** | **0.8518** |

Earlier, we implemented models that can serve as benchmarks for comparison in this work. These previous models were applied to the same dataset and provide a foundation for evaluating the performance of our new deep learning approaches. The results from these models allow us to objectively assess the progress and improvements made with the new algorithms.

The Table 1 and fig 12 presents a comparison of the performance of the earlier models (Moreh et al., 2024) with the current results. This overview provides a clear comparison of the different approaches and highlights the improvements achieved through the application of more advanced deep learning techniques. Our method provides better performance on DSC and IoU metrics, suggesting that the proposed model is not only able to detect the crack with high accuracy but also localize it better. A higher IoU score also suggests that the predicted crack region and the ground truth region have a high overlap, meaning the model is drawing the crack boundary closer to where the crack actually exists. With better DSC and IoU, the model is also less likely to make mistakes. It won't detect cracks where there are no cracks present (false positives) or miss cracks that should be detected (false negatives). This is important for real-world applications like structural health monitoring where accuracy is critical.

## REFERENCES

Fatahlla Moreh, Hao Lyu, Zarghaam Haider Rizvi, and Frank Wuttke. Deep neural networks for crack detection inside structures. *Scientific Reports*, 14(1):4439, Feb 2024. ISSN 2045-2322. doi: 10.1038/s41598-024-54494-y. URL https://doi.org/10.1038/s41598-024-54494-y.

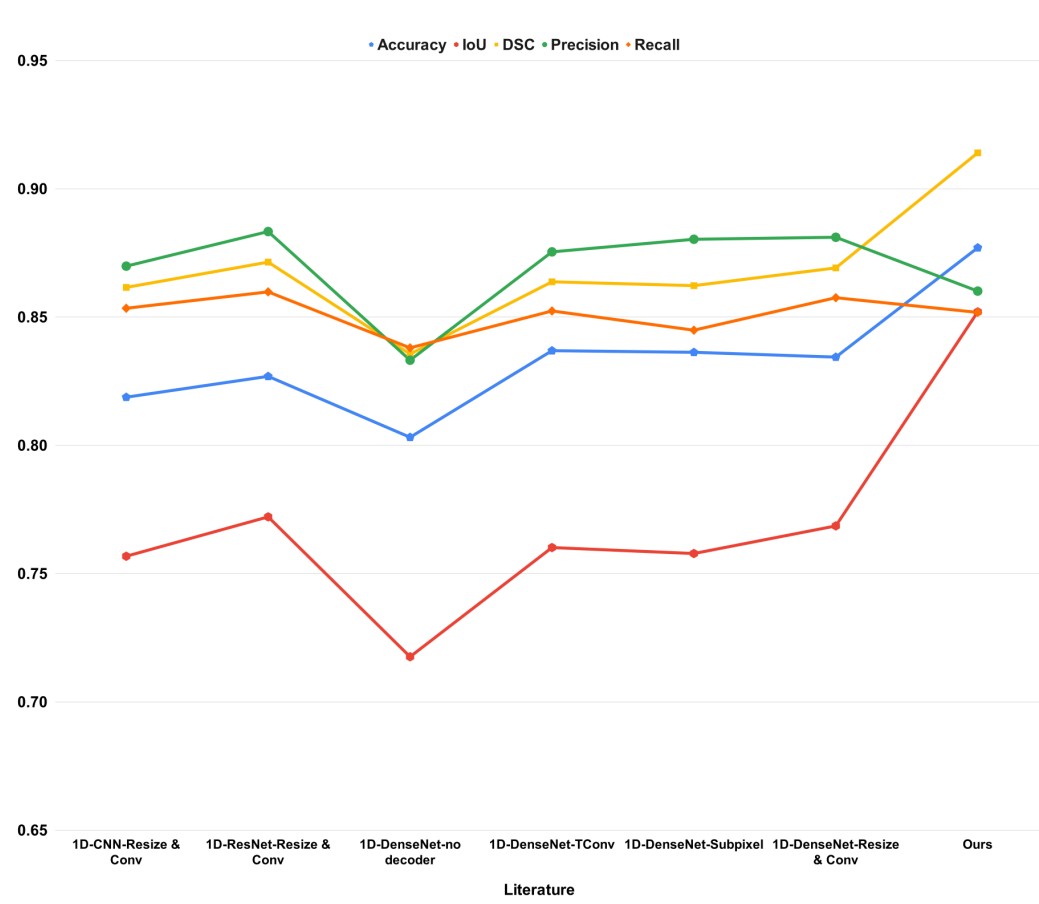

Figure 12: Comparison against previous works on the same dataset.