# OpenReview forum: "MicroCrackAttentionNeXt: Advancing Microcrack Detection in Wave Field Analysis Using Deep Neural Networks through Feature Visualization."
_ICLR.cc/2025/Conference — ICLR 2025 Conference Withdrawn Submission_

### Official Review · Reviewer_KSWk · 2024-10-15

**Soundness:** 2
**Presentation:** 2
**Contribution:** 1
**Rating:** 1
**Confidence:** 4

**Summary:**

This work presents a deep neural network architecture that is designed to output segmentation maps of micro cracks. Pixels that represent micro crack are scarce in the dataset (5%), so it is essential to deal with the class imbalance. This works extends the previous work 1D-DenseNet and presents improved performance results. In addition it offers MDA visualizations for its inner layers.

**Strengths:**

The paper written clearly and is easy to follow.
It offers an accuracy improvement from 83.68% to 86.85%.

**Weaknesses:**

1) The work is compared to only a single architectural alternative on a single dataset. The only comparison made is with the work that this study is heavily based on, and even this comparison is incomplete. What are the accuracies, DSC, and IoU in comparison to the other work?
Please present a table or figure comparing accuracies, DSC, and IoU scores between the proposed method and the baseline.
Instead of extending the evaluation to different datasets or comparing it with other techniques, only a few ablation experiments were presented, focusing on different losses or activation functions. The authors are encouraged to evaluate/consider recent segmentation techniques or, at the very least, explain why recent architectures, such as those based on transformers, are excluded from the comparison. Recent segmentation techniques may have a much greater performance impact compared to the improvements derived from investigating different activation/loss functions.
For example, will an adaptation of SAM2 (or any other recent alternative) for your kind of data, might work?

1.5) It is stated in the related work that this study extends 1D-DenseNet and is heavily influenced by it, but it is unclear what the specific similarities are and what the extensions consist of. Additionally, it is not clear which modifications lead to the observed improvements.
It might be helpful if the authors provide a specific section or table that clearly outlines the similarities and differences between their proposed model and 1D-DenseNet, as well as explicitly linking each modification to its impact on performance.

2) While this work might contribute to scientific progress in the field of materials inspection, I couldn't identify any novelty in the field of machine learning. The work employs well-known components, such as convolutional layers and self-attention layers, in an architecture that is largely based on a previous work. It suggests using established loss functions and activation functions.

3) The use of MDA is not well explained. I don’t understand what contribution the MDA visualizations make. Specifically, how do they help to understand the model's inner workings or how it performs compared to other alternatives? Additionally, MDA evaluates the model based on another "black-box" DNN algorithm. Instead, a more concise approach would be to base the explanation on well-established metrics (such as DSC or accuracy) or straightforward visualizations from the model, such as attention maps, to demonstrate semantic understanding. From my perspective, simply demonstrating improved DSC or accuracy is more convincing for evaluating a segmentation model. This contrasts with what is stated in lines 77-78.

Technical Issues:

Line 205: "Figure X" needs to be specified.

A citation or definition for Squeeze-and-Excitation layers would be helpful.

Lines 50-53: The soundness of the claim is unclear. Your architecture also includes residual connections, and it’s not necessarily the case that UNet’s reliance on residual connections is the reason for its underperformance compared to attention layers.

Lines 66-69: The loss function description feels unnatural and could be presented more clearly.

Lines 340-347: I expected to see Focal Loss mentioned somewhere here.

**Questions:**

1) I did not understand whether 1D or 2D convolutional layers were used. If it is 1D, I don't understand the reason as the spatial data is 2D.
If it is 2D, 1D is written in the conclusion.

2) What are the performance reports in the related work (lines 133, 143, 148)? Were all these tested on the same dataset and settings as this work? If so, you should present these comparison in the experiments section.

---

### Official Review · Reviewer_s5c7 · 2024-10-31

**Soundness:** 3
**Presentation:** 2
**Contribution:** 1
**Rating:** 3
**Confidence:** 5

**Summary:**

This study presents a network for crack detection using observed seismic data. An attention mechanism is incorporated to effectively map spatio-temporal data to spatial data. They compare accuracies across different loss functions for various crack sizes. The results demonstrate that the proposed method achieves satisfactory performance.

**Strengths:**

They considered the complex relationship between spatio-temporal seismic data to spatial detection result.

**Weaknesses:**

1. The experiments are insufficient, lacking an ablation study and visual comparisons.
2. The novelty is limited, as this work merely applies an attention-based network to crack detection.
3. No field tests are conducted, which raises concerns about the generalizability of the findings.
4. The dataset settings are unclear.

**Questions:**

How is the training dataset prepared? Is it collected from real data or generated synthetically?

---

> ### Author Response · Authors · 2024-12-02
> **Reply to reviewers comments**
>
> Weaknesses:
>
> Insufficient Experiments and Lack of Ablation Study: Addressed in the supplementary section.
>
> Limited Novelty: While the attention mechanism and network architecture may appear as incremental, the novelty lies in their application to our custom unique dataset of spatio-temporal seismic wave data, which is significantly different from traditional image-based datasets. The Adaptive Feature Reutilization Block and the combined use of activation functions and loss metrics are custom-designed for this dataset to address its non trivial modality and severe class imbalance.
>
> No Field Tests and Generalizability Concerns: We acknowledge that field tests are crucial to validating the model's generalizability. However, the current study focuses on demonstrating the feasibility of using seismic wave field data for crack detection through simulations. The dataset used was specifically designed to include varying crack sizes, orientations, and noise levels to mimic real-world scenarios.
>
> Unclear Dataset Settings: The dataset used in this study was synthetically generated using numerical simulations of wave propagation in homogeneous plates with cracks, as described in Section 3.1. The spatial and temporal dimensions of the wave data were recorded over 2000 timesteps using a 9x9 sensor grid.
>
> Questions:
> How is the Training Dataset Prepared: The dataset used in this study is synthetically generated through numerical simulations of seismic wave propagation in a homogeneous 2D plates, where each plate is modeled with lattice particles that share consistent properties, such as density and Young’s Modulus. The modeling of structural systems is achieved using Voronoi-Delaunay meshing algorithms within the Lattice Element Method (LEM).  Cracks of varying sizes and orientations were introduced into the plate, and a simulated force was applied to induce wave propagation. The resulting displacements in both x and y directions were recorded over 2000 time steps using a 9x9 sensor grid as described in Section 3.1.
> This approach ensures precise control over the dataset's characteristics, such as crack size, orientation, and location, which are challenging to achieve with real-world data. Future work will involve extending the model to real-world datasets by incorporating noise and variability observed in field measurements to validate its generalizability.

---

> > ### Comment · Reviewer_s5c7 · 2024-12-03
> >
> > This work has several issues and lacks novelty, as highlighted by other reviewers. I maintain my initial ranking.

---

### Official Review · Reviewer_ypNZ · 2024-11-02

**Soundness:** 1
**Presentation:** 2
**Contribution:** 1
**Rating:** 1
**Confidence:** 4

**Summary:**

In this paper, authors propose a MicroCrackAttentionNeXt model for the micro crack detection, and utilize a Manifold Discovery and Analysis (MDA) method to visualize the learned feature of the network.

**Strengths:**

The structure of this paper is clear.

**Weaknesses:**

This paper only applies an existing MDA method for the crack detection, and compares the performances of model with different activation and loss functions. It is lack of innovation. Additionally, the quantitative comparisons with other existing crack detection models are not provided.

**Questions:**

1. In Abstract, the motivation and innovation are not mentioned.
2. The advantage and disadvantage of the existing related works are not analyzed comprehensively. So, the motivation of this paper is not clear.
3. What is Figure X in Page 4? What is the relationship of MicroCrackAttentionNeXt model and Squeeze-and-Excitation layers in Page 5? Moreover, Figure 3-5 are not described in the paper.
4. The evaluation metrics are very important, but they are not mentioned in this paper. Since the quantitative detection results of the proposed MicroCrackAttentionNeXt and other state-of-the-art crack detection models are not given, it is difficult to define the contribution of this paper.
5. There are some grammatical mistakes, such as “The dataset presents a substantial class imbalance, with crack pixels constituting an average of only 5% of the total pixels per sample, this extreme class imbalance poses a challenge for deep learning models with the different micro scale cracks, as the network can be biased toward predicting the majority class, generally leading to poor detection accuracy.”

---

> ### Author Response · Authors · 2024-12-01
> **Response to Reviewer Comments**
>
> Weaknesses:
>
> Lack of Innovation: The proposed model integrates existing techniques like the MDA method, but the innovation lies in crafting a model capable of detecting and segmenting such minute cracks from a huge spatio-temporal wave field data. This dataset poses significant challenges, such as high dimensionality and severe class imbalance, which are not typically addressed in the literature. Additionally, the introduction of Adaptive Feature Reutilization Blocks and the optimized combination of activation functions and loss metrics provide novel contributions tailored to this domain. Our work uses wave propagation through materials to enable efficient identification of flaws without the need for expert monitoring during the inspection process, enhancing the reliability of detection of structural cracks in high profile assets.
>
> Quantitative Comparisons with Existing Models: Direct comparisons with other state-of-the-art crack detection models were not provided, as these models are designed for image-based datasets and cannot be directly applied to the numerical wave field data used in this study. However, we have benchmarked the proposed model against previously published methods specifically designed for this dataset in the supplementary materials. Future work will involve creating additional benchmarks to evaluate the performance of external models on our dataset.
>
> Evaluation Metrics: Addressed in supplementary section.
>
> Grammatical Errors and Presentation: We acknowledge the grammatical errors and the need for better presentation. They have been addressed in the revised version.
>
> Questions:
>
> Motivation and Innovation in Abstract: Addressed in the revised version.
>
> Related Work and Motivation: Addressed in the related works section, paragraph 2.
>
> Figures and Descriptions: Figure X on Page 4: Addressed in the revision.
>
> Evaluation Metrics: Addressed in supplementary section.
>
> Grammatical Mistake: We do not see any grammatical mistakes in those lines, but still have been addressed in the revision.

---

### Official Review · Reviewer_w1Ui · 2024-11-03

**Soundness:** 3
**Presentation:** 2
**Contribution:** 2
**Rating:** 3
**Confidence:** 3

**Summary:**

This paper proposes the MicroCrackAttentionNeXt, a deep neural network model designed to enhance microcrack detection in materials using wave field data. Building upon SpAsE-Net, this model introduces an asymmetric encoder-decoder structure and leverages attention mechanisms to better capture spatio-temporal interactions critical for microcrack detection. Key elements include various activation functions and loss metrics, evaluated through the Manifold Discovery and Analysis (MDA) approach for feature visualization. The paper demonstrates that the combination of the Gaussian Error Linear Unit (GeLU) activation and Combined Weighted Dice Loss (CWDL) achieved optimal performance, resulting in an accuracy of 86.85%.

**Strengths:**

1.  The asymmetric encoder-decoder with attention mechanisms offers a promising approach to tackle the complexity of spatio-temporal data in microcrack detection.

2. The exploration of different activation functions and loss metrics provides valuable insights into model optimization for class-imbalanced data.

3. The application of MDA to visualize feature representations in higher dimensions is well-executed, giving a qualitative assessment of model behavior across layers and activation functions.

**Weaknesses:**

1. **Dataset and Class Imbalance**: The paper notes severe class imbalance, which could impact the generalizability of results. Although methods are employed to mitigate this, it remains a limitation without further exploration into data augmentation or synthetic generation techniques.

2.  **Baseline Models**: While the paper references prior models, including SpAsE-Net, direct quantitative comparisons against other state-of-the-art microcrack detection models are limited, which may hinder assessing MicroCrackAttentionNeXt's performance gains.

3. **Resolution of Output Segmentation**: The paper mentions that the output segmentation suffers from low resolution, which may limit its applicability in scenarios demanding high-resolution segmentation for precise crack localization.

4. **Scalability and Computational Efficiency**: Although the model incorporates temporal downsampling to manage data size, the practical scalability of MicroCrackAttentionNeXt to larger datasets or higher-resolution scenarios could be further discussed.

**Questions:**

1. **Model Generalizability Across Varying Conditions**: The dataset's severe class imbalance and limited temporal resolution are acknowledged but not adequately addressed. How can the authors justify the model’s generalizability in detecting microcracks under different material compositions or wave propagation scenarios, especially given the narrow dataset? Could this limit the model's application in real-world, diverse settings?

2. **Comparative Baselines**: Although the paper positions MicroCrackAttentionNeXt as an improvement over SpAsE-Net, it lacks direct quantitative comparison with a broader range of state-of-the-art models in microcrack detection. Without such comparisons, how can the authors substantiate claims of improved accuracy or efficiency?

3.  **Low-Resolution Segmentation**: The paper concedes that the segmentation output’s low resolution could lead to loss of detail in crack localization. Given this limitation, how does the model ensure precise identification of microcracks, particularly those close to the resolution limit? Could this restriction render the model ineffective for critical applications requiring high localization accuracy?

4. **Evaluation Metrics**: The paper predominantly relies on the accuracy and Dice Similarity Coefficient (DSC), but these may not fully capture the model’s capability in highly imbalanced, nuanced detection tasks. Why were more detailed metrics, such as precision-recall curves or area under the ROC curve (AUC), not included to provide a more comprehensive evaluation? Furthermore, was any statistical validation (e.g., confidence intervals) performed to ensure the robustness of the reported performance metrics?

---

> ### Author Response · Authors · 2024-12-01
> **Response to Reviewer Comments**
>
> Weaknesses:
>
> Dataset and Class Imbalance: The dataset's inherent class imbalance was addressed through loss functions such as the Combined Weighted Dice Loss, which directly emphasizes the minority class (crack regions). While advanced data augmentation techniques or synthetic generation were not included in this iteration, it is rather not straight forward to include an off the shelf data augmentation techniques to this unique data modality.
>
> Baseline Models: As mentioned in the above comment, an apples to apples comparison cannot be made with other models, given the uniqueness of the data that we are dealing with. In spite of this, the supplementary section has been populated with some previous benchmark models on the same data.
>
> Resolution of Output Segmentation: The output resolution is indeed a limitation, driven by the need to balance computational complexity and model performance. However, the segmentation accuracy indicates that the model effectively captures key crack features. Assessing the current literature, there are only few works that are currently detecting cracks in such data modality, let alone segment them. In this work, we focused primarily on the encoder section, in future work we plan to explore super-resolution techniques, to improve the spatial detail of segmentation outputs.
>
> Scalability and Computational Efficiency: The scalability of the model was addressed through the use of temporal downsampling and lightweight attention mechanisms. Preliminary results suggest that the model performs efficiently on the current dataset size, but scaling to higher resolutions or larger datasets remains a challenge. We plan to optimize the architecture further, potentially by leveraging parameter-sharing techniques and lightweight attention modules.
>
> Questions:
>
> Model Generalizability Across Varying Conditions: The dataset used was specifically designed to include varying crack sizes, orientations, and noise levels to mimic real-world scenarios. The model's generalizability is supported by its robustness to class imbalance and noise, as demonstrated through its performance on synthetic data. Future work will involve validation on real-world datasets with diverse material compositions and wave propagation patterns to strengthen generalizability claims.
>
> Comparative Baselines: As addressed in weaknesses section, a direct comparison with other state of the art vanilla models is not possible, simply because they cannot be applied out of the box for this dataset. However, we have provided a comparison with previous state of the art models on the same dataset.
>
> Low-Resolution Segmentation: Addressed in the weakness section.
>
> Evaluation Metrics: Addressed in the supplementary section.

---

### Official Review · Reviewer_6hKS · 2024-11-08

**Soundness:** 3
**Presentation:** 2
**Contribution:** 2
**Rating:** 3
**Confidence:** 4

**Summary:**

The paper introduces MicroCrackAttentionNeXt, an advanced deep learning model designed to enhance microcrack detection in structural materials using wave field analysis. Traditional CNNs struggle with the complex spatio-temporal patterns and severe class imbalance (cracks constitute only 5% of data). The model uses an asymmetric encoder-decoder architecture with attention mechanisms, inspired by existing structures such as SpASe-Net, but optimized for micro-scale feature detection. The authors also explore the impact of various activation functions and loss strategies through Manifold Discovery and Analysis (MDA), aiming to improve feature separability and reduce overfitting. The proposed model achieves a significant accuracy of 86.85%, outperforming benchmark models in microcrack segmentation.

**Strengths:**

1. **Soundness of Claims:**
   - The study provides strong empirical evidence for the model's performance, demonstrated through experiments comparing *MicroCrackAttentionNeXt* against established benchmarks like 1D-DenseNet. The use of multiple activation and loss function combinations showcases the robustness of the approach.
   - The application of MDA for qualitative analysis adds depth to the understanding of the learned representations, illustrating the model's ability to separate complex features effectively.
   - The theoretical foundation, leveraging attention mechanisms and hierarchical feature extraction, is well-grounded in modern deep learning literature, enhancing the reliability of the results.

2. **Significance:**
   - The model addresses a critical problem in the field of structural health monitoring, where microcrack detection is vital for preventing catastrophic failures. The real-world implications of this work extend to various engineering applications, making it highly impactful.
   - The research introduces a nuanced solution to the issue of class imbalance, a common challenge in segmentation tasks, by experimenting with different loss functions tailored to emphasize minority classes.
   - The study's contribution lies in the integration of MDA, offering a new perspective on model interpretability and feature visualization, which can be valuable for future research in deep learning-based structural analysis.

3. **Novelty:**
   - The paper presents a novel architecture by combining a tailored asymmetric encoder-decoder design with specialized attention modules, enhancing the detection of small, complex features like microcracks.
   - The comprehensive analysis of activation functions, rarely explored in-depth in this context, brings a fresh approach to optimizing neural network performance for this task.
   - The proposed use of manifold analysis for qualitative feature evaluation is innovative and provides new insights into the model's inner workings, setting it apart from traditional performance metrics.

**Weaknesses:**

1. **Soundness of Claims:**
   - While the empirical results are compelling, the paper could benefit from a more extensive comparison with a broader range of models, including state-of-the-art transformer-based architectures, to validate the superiority of *MicroCrackAttentionNeXt*.
   - The theoretical justification for the chosen architecture and specific configurations, such as the kernel sizes and pooling layers, lacks detailed mathematical support or ablation studies to isolate the effects of these choices.
   - The MDA analysis, though informative, appears somewhat qualitative; incorporating more quantitative measures to assess feature separability could strengthen the argument.

2. **Significance:**
   - The model's performance improvement, while notable, is not groundbreaking when considering the field's rapid advancements. An increase from previous benchmarks may not justify the added architectural complexity.
   - The study's reliance on synthetic data for training and validation could limit its applicability in real-world scenarios, as the dynamics of wave propagation in laboratory settings may differ from those in practical engineering contexts.
   - There is a lack of discussion on how the proposed approach scales with larger datasets or more complex wave forms, which could limit its feasibility in extensive industrial applications.

3. **Novelty:**
   - Although the architecture is tailored for this application, many components are adaptations of existing methods, such as attention mechanisms and encoder-decoder networks. The paper does not significantly deviate from established deep learning paradigms.
   - The paper could explore more groundbreaking methodologies, such as incorporating graph-based networks for modeling wave propagation more naturally.
   - The novelty of using MDA is limited by the fact that it only provides interpretability benefits without contributing directly to performance enhancement.

**Questions:**

1. How does the model perform when tested on real-world datasets compared to synthetic wave field data?
2. Are there specific scenarios or material properties where *MicroCrackAttentionNeXt* performs poorly, and how can these be addressed?
3. Can the proposed model handle various noise levels in wave data, which are common in real-world applications?
4. What is the computational efficiency of the model during training and inference compared to simpler architectures?
5. How would the model's performance vary if it were extended to handle 3D wave propagation data?

**Details Of Ethics Concerns:**

#### A. Major Issues

1. **Dependence on Synthetic Data:**
   - The experiments heavily rely on synthetic wave field data, which may not accurately reflect the conditions encountered in real-world microcrack detection. This limits the model's generalizability and real-world applicability.
2. **Class Imbalance Mitigation:**
   - Although the paper addresses class imbalance using various loss functions, it does not explore advanced data augmentation techniques or other balancing strategies, which could further improve performance.
3. **Scalability Concerns:**
   - The architectural complexity, including attention mechanisms and multiple down-sampling layers, raises concerns about the model's scalability and efficiency on larger datasets or in deployment scenarios.
4. **Limited Activation Function Analysis:**
   - The activation function analysis, while comprehensive, could have explored novel activations beyond the commonly used ReLU, SELU, GELU, and ELU variants to potentially uncover better-performing alternatives.
5. **Inadequate Ablation Study:**
   - The study lacks a detailed ablation analysis to isolate the impact of each architectural component, such as SE modules or specific down-sampling strategies, reducing the interpretability of the model's design choices.

#### B. Minor Issues

1. **Insufficient Hyperparameter Justification:**
   - The choice of hyperparameters, such as the learning rate and pooling sizes, is not well-justified, which could impact reproducibility.
2. **Limited Discussion on Training Dynamics:**
   - The paper does not discuss the model's convergence behavior or challenges faced during training, such as instability or overfitting.
3. **Visual Representation Limitations:**
   - Figures and visualizations of the MDA analysis could be clearer, especially in distinguishing the representations of trained versus untrained models.

---

### C. Recommendations

1. Improve the analysis by incorporating real-world datasets to validate the model's performance and robustness.
2. Provide a more extensive ablation study to clarify the importance of each architectural component and hyperparameter setting.
3. Clarify visual representations and ensure the figures clearly depict the model's feature separability improvements.

---

> ### Author Response · Authors · 2024-11-28
>
> Weaknesses and Questions Addressed:
>
> Broader Comparison with Transformer-Based Architectures: We agree that incorporating state-of-the-art transformer-based models could further validate our approach. However, an apples to apples comparison cannot be made for transformer based models given the uniqueness of the data that we are dealing with. In spite of this, the supplementary section has been populated with some previous benchmark models on the same data.
>
> Theoretical Justifications and Ablation Studies: We acknowledge that a more detailed ablation study and theoretical justification of certain architectural choices, such as kernel sizes and pooling strategies, could strengthen our arguments. These were omitted due to space constraints but will be included in supplementary material or future publications.
>
> Quantitative MDA Analysis: While MDA's qualitative insights have been instrumental in understanding feature separability, we recognize the value of incorporating quantitative measures like silhouette scores or inter-class distance metrics. We plan to integrate these metrics into future revisions.
>
> Performance on Real-World Data: Our reliance on synthetic data stems from the lack of publicly available real-world datasets for this specific problem. However, the synthetic data were generated using advanced numerical simulations designed to closely mimic real-world conditions.
>
> Handling Noise and 3D Data: The data used in our study was generated to include simulated noise, designed to closely mimic real-world conditions. This ensures the model's robustness to noise. While the current work focuses on 2D wave propagation, extending the model to handle 3D wave propagation data is an exciting avenue we plan to explore in future studies.
>
> Computational Efficiency: The computational efficiency of our model is comparable to standard attention-based encoder-decoder networks. While the added complexity introduces a marginal increase in training time, it does not significantly impact inference efficiency. A quantitative result is presented here - The adaptive 1D densenet take 200 epochs to train and reach a slightly lower accuracy score than the proposed model being trained for 50 epochs.
>
> Activation Function Exploration: We appreciate the suggestion to explore novel activation functions beyond common variants. While this was beyond the scope of our initial study, we are intrigued by emerging activations like Swish and Mish and will evaluate their applicability in future work.
>
> Ethics Concerns:
>
> Dependence on Synthetic Data: While synthetic data is an inherent limitation, our methodology is designed to be transferable to real-world applications. Initial validations indicate that our model captures fundamental patterns relevant to microcrack detection.
>
> Class Imbalance Mitigation: We agree that advanced augmentation techniques, such as GAN-based synthetic crack generation, could further address class imbalance. We are not yet sure if these techniques are feasible for the nature of the dataset that we are dealing with.

---

### Note · Authors · 2025-01-12

I have read and agree with the venue's withdrawal policy on behalf of myself and my co-authors.